# Treatment Strategies for ARID1A-Deficient Ovarian Clear Cell Carcinoma

**DOI:** 10.3390/cancers13081769

**Published:** 2021-04-07

**Authors:** Kazuaki Takahashi, Masataka Takenaka, Aikou Okamoto, David D. L. Bowtell, Takashi Kohno

**Affiliations:** 1Department of Obstetrics and Gynecology, Jikei University School of Medicine, Tokyo 105-8461, Japan; Kazuaki.Takahashi@petermac.org (K.T.); tt10mm10@jikei.ac.jp (M.T.); aikou@jikei.ac.jp (A.O.); 2Peter MacCallum Cancer Centre, Melbourne, VIC 3000, Australia; david.bowtell@petermac.org; 3Division of Genome Biology, National Cancer Center Research Institute, Tokyo 104-0045, Japan; 4Sir Peter MacCallum Department of Oncology, The University of Melbourne, Parkville, VIC 3010, Australia

**Keywords:** ovarian clear cell carcinoma, ARID1A, synthetic lethality, gemcitabine, molecular targeted therapy, precision medicine

## Abstract

**Simple Summary:**

Deleterious mutations in SWI/SNF chromatin remodeling genes, such as *ARID1A*, are present in more than 50% of cases of ovarian clear cell carcinoma (OCCC), a histological subtype of ovarian cancer prevalent in Asian countries. To efficiently treat OCCC, which is refractory to conventional platinum-based chemotherapy, several therapeutic strategies based on SWI/SNF deficiency have been proposed, including gemcitabine-based chemotherapy, synthetic lethal therapy, and immune checkpoint blockade therapy. Implementation of these strategies would improve the prognosis of patients with this disease.

**Abstract:**

Ovarian clear cell carcinoma (OCCC) is a histological subtype of ovarian cancer that is more frequent in Asian countries (~25% of ovarian cancers) than in US/European countries (less than 10%). OCCC is refractory to conventional platinum-based chemotherapy, which is effective against high-grade serous carcinoma (HGSC), a major histological subtype of ovarian cancer. Notably, deleterious mutations in SWI/SNF chromatin remodeling genes, such as *ARID1A*, are common in OCCC but rare in HGSC. Because this complex regulates multiple cellular processes, including transcription and DNA repair, molecularly targeted therapies that exploit the consequences of SWI/SNF deficiency may have clinical efficacy against OCCC. Three such strategies have been proposed to date: prioritizing a gemcitabine-based chemotherapeutic regimen, synthetic lethal therapy targeting vulnerabilities conferred by SWI/SNF deficiency, and immune checkpoint blockade therapy that exploits the high mutational burden of *ARID1A*-deficient tumor. Thus, ARID1A deficiency has potential as a biomarker for precision medicine of ovarian cancer.

## 1. Introduction

Ovarian clear cell carcinoma (OCCC) is a histological subtype of ovarian cancer that constitutes 25% of ovarian cancers in Asian countries, but less than 10% of ovarian cancers in US and European countries [1,2,3]. OCCC is more refractory to conventional platinum-based chemotherapy than other major histological types of ovarian cancer, such as high-grade serous carcinoma (HGSC) [4]; the response rate in OCCC is 11–56%, whereas that in HGSC is about 80% [5,6,7]. Because OCCC is rare in US and European countries, OCCC cases have not been actively enrolled into clinical trials, and clinical trials specifically targeting OCCC have been highly limited [8]. Consequently, effective treatment strategies for OCCC (i.e., precision medicine) have not yet been established [4,9].

OCCC is characterized by genetic alterations distinct from those found in HGSC, including frequent deficiency of genes encoding subunit proteins of the SWI/SNF chromatin remodeling complexes. The nucleosome, the basic unit of chromatin, is composed of 146 pairs of DNA bases wrapped around histone protein octamers. Nucleosomes prevent the binding of transcription factors and histone modifiers in the nucleus. Chromatin remodeling complexes regulate gene expression, DNA replication and repair, and cell division through changes in chromatin structure. The SWI/SNF complexes, which comprise tens of subunit proteins, obtain energy from ATP hydrolysis to cause nucleosome sliding, exposing specific regions of DNA and allowing interaction with histone modifiers [10]. The *ARID1A* gene, which encodes the BAF250A/ARID1A protein, is the most frequently mutated SWI/SNF subunit gene in OCCC, although several other subunit genes, including *SMARCA4* and *ARID1B*, are also mutated [11]. *ARID1A* mutations, most of which are deleterious, are detected in about 50% of OCCCs, and loss of BAF250A/ARID1A protein, which functions as a regulatory subunit of the SWI/SNF complex, is observed at a similar frequency [12,13]. Interestingly, loss of BAF250A/ARID1A protein expression is observed not only in homozygous but also in heterozygous mutants. A previous study reported the post-transcriptional/translational effects of *ARID1A* mutation [13]. Several studies reported that knockout of the *ARID1A* gene impairs transcriptional and DNA repair activities within cells [14,15]; therefore, the function of the SWI/SNF complex is (at least partially) lost in half of OCCC cases. In this review, we focus on the properties of OCCC and possible precision medicine for this cancer from the standpoint of deficiency in SWI/SNF-mediated chromatin remodeling.

## 2. Inter-Ethnic Differences in the Prevalence of OCCC

The proportion of OCCC among the four major histological types of epithelial ovarian cancer is higher in Asia (about 25%) than in Europe and the US (less than 10%) [1,2,3]. The proportions in Japan, the US, and Australia are shown in Figure 1a. The data from Japan and the US are from the Japan Society of Obstetrics and Gynecology tumor registry database and the Surveillance, Epidemiology, and End Results (SEER) program (2002–2015) [16], respectively, whereas the data from Australia is from a nationwide epidemiological survey conducted by the Australian Ovarian Cancer Study between 2001 and 2005 [17,18,19]. The Japanese cohort included a higher fraction (27%) of OCCC than the US (8%) and Australian (8%) cohorts, respectively.

We estimated the numbers of new OCCC cases per year based on the projected number of ovarian cancer patients (PNP) in Japan, the US, and Australia in 2020, and the proportion of OCCC among all ovarian cancers, given above. The number of patients with OCCC, which we estimated by multiplying the PNP by the proportion of OCCC in each cohort, is shown in Figure 1b. The number of overall ovarian cancer cases is smaller in Japan (13.4 K) than in the US (21.8 K), whereas the number of OCCC cases is likely to be 2-fold greater in Japan (3.6 K) than in the US (1.7 K). This estimate indicates that not only the proportion but also the absolute number of OCCC patients is larger in Asia (using Japan as a representative) than in the US. The reasons for such inter-ethnic differences are unknown. However, these calculations validate the idea that OCCC is a particularly problematic disease in Japan and other Asian countries.

## 3. *ARID1A* and Other SWI/SNF Gene Alterations in OCCC

Multiple genes encoding subunits of the SWI/SNF complexes, including *ARID1A*, *SMARCA4*, *PBRM1*, and *SMARCB1*, are mutated in ~20% of all cancers [20,21]. In particular, several pan-cancer genome-wide studies repeatedly identified *ARID1A* as one of the top 10 most frequently mutated genes [22,23,24]. In particular, OCCC is frequently associated with *ARID1A* mutations [12,13], while other SWI/SNF subunit genes, such as *ARID1B* and *SMARCA4*, are also mutated in a subset of OCCC [11]. Mutation frequencies of SWI/SNF subunit genes in the project GENIE database (v8.1), which accumulated data from gene profiling tests performed in daily oncology practice in the US [25], are shown in Figure 2a,b. *ARID1A* is the most frequently mutated SWI/SNF gene, followed by *ARID1B*, *SMARCA4*, *ARID2,* and *PBRM1*. The mutation frequencies of these genes according to tumor type are shown in Figure 2c–g. The distribution of mutations in genes related to SWI/SNF complexes is not random; rather, such mutations are more common in certain cancer types [26]. Evidently, frequent *ARID1A* mutation is a genetic property of OCCC. *ARID1A* mutations are detected in adjacent endometriotic lesions of OCCC, but not in distant endometriosis in the same patient, indicating that OCCC arises from endometriosis, and that *ARID1A* functions as a major tumor suppressor during development of OCCC [13,27,28,29].

*ARID1A* is also frequently mutated in uterine endometrial and endometrioid ovarian cancers; therefore, aberrations in this gene are likely to be frequently involved in malignancies of the female genital tract. In addition, *SMARCA4*, which is frequently mutated in poorly differentiated non-small cell lung cancer (Figure 2g), is also mutated in a subset (about 10%) of OCCC. *SMARCA4* mutations are also extremely frequent (almost ubiquitous) in small cell carcinoma of the ovary of hypercalcemic type (SCCOHT), occurring in about 98% of such cases. Somatic and germline *SMARCA4* mutations drive development of this disease [30,31]. Therefore, deficiency of SWI/SNF chromatin remodeling contributes to ovarian cancer development in several ways.

Figure 3a summarizes the frequencies of *ARID1A* mutations and loss of BAF250A/ARID1A protein expression in OCCC in studies to date [12,13,32,33,34,35,36,37]. In particular, many immunohistochemical studies to examine loss of BAF250A expression in OCCC tissues have been performed [38,39,40]. Loss of BAF250A, reflected by lack of nuclear staining for this protein, is observed frequently in OCCC cells. The frequency of *ARID1A* mutations in OCCC is ~60% in the US, Canada, and Japan, and 40% in Australia, and the frequencies of loss of BAF250A expression are equal to or a bit lower than the mutation frequencies. These data indicate that a significant fraction of *ARID1A* mutations is associated with loss of expression of the gene product. Consistent with this, *ARID1A* mutations observed in OCCC in the Project GENIE Cohort consist mostly (>90%) of truncating mutations (Figure 3b, upper) that are dispersed throughout the coding sequence. *ARID1A* contains two DNA-binding domains, the ARID domain (AT-rich interacting domain) and the C-terminal domain, which plays a critical role in promoting transcription [41]. Most truncation mutations remove the C-terminal domain, whereas missense mutations affecting the glycine at position 2087 decrease the stability of the ARID1A protein [42]. This predominance of truncation mutations and missense mutations at Gly2087 is also observed in a wide range of cancers other than OCCC. Thus, deleterious *ARID1A* mutations are a common feature of OCCC and other cancers.

The pathogenic roles of ARID1A deficiency during OCCC development were recently elucidated. Indeed, somatic mutations in the *ARID1A* and *PIK3CA* genes are detected in benign endometriosis [29,43]. Concordantly, a study using a mouse model demonstrated that *ARID1A* deficiency, in concert with an oncogenic *PIK3CA* mutation, promotes OCCC formation in vivo by enhancing inflammatory cytokine signaling [44]. In addition, another study demonstrated that alterations in both the ARID1A and PI3-Kinase (PI3K) pathways promote epithelial trans-differentiation and invasion [45]. The epigenetic role of *ARID1A* was also revealed in a pre-OCCC model system: ARID1A prevents super-enhancer hyperactivation, which leads to enhanced migratory properties exhibited by pre-OCCC cells [46]. In addition, a common role shared by ARID1A and another SWI/SNF factor, SMARCA4/BRG1, maintains the integrity of the endometrial epithelium [47]. Thus, ARID1A and other SWI/SNF factors are likely to function epigenetically as a tumor suppressor for the development of ovarian cancer. 

## 4. Therapeutic Strategies for OCCC Based on the Phenotypes of ARID1A Deficiency

The BAF250A/ARID1A protein functions as a regulatory subunit of the SWI/SNF complex, which regulates multiple cellular processes, including transcription and DNA repair. Therefore, it is possible that cancer cells deficient in ARID1A or other SWI/SNF subunits share properties that are absent from noncancerous cells, and that some of those properties create vulnerabilities [10]. Targeting this vulnerability with anti-cancer drugs represents a potentially effective strategy for treating OCCC. Indeed, several studies to date have proposed such approaches, which are categorized into three groups.

### 4.1. Prioritizing Gemcitabine-Based Chemotherapeutic Regimens

Gemcitabine is a deoxycytidine analogue that inhibits ribonucleoside reductase, resulting in depletion of deoxyribonucleotide pools required for DNA synthesis. Gemcitabine is often used in late lines of treatment for ovarian cancer after platinum-resistant recurrence; however, retrospective studies have shown that gemcitabine is especially effective against OCCC [48,49,50]. Notably in this regard, we demonstrated that knockout of *ARID1A* increases the sensitivity of OCCC cells to gemcitabine by approximately 100-fold. Consistent with this, ARID1A-deficient cases of OCCC exhibited significantly longer progression-free survival after gemcitabine treatment than ARID1A-proficient cases [51]. The mechanisms underlying this phenomenon are unknown, but it seems likely that patients with ARID1A-deficient OCCC would benefit from treatment with gemcitabine.

### 4.2. Synthetic Lethal Therapy Targeting Vulnerabilities Conferred by ARID1A Deficiency

Loss-of-function mutations in the *BRCA1* and *BRCA2* genes have opened the prospect of developing new synthetic lethal therapies based on PARP inhibitors [52]. However, these therapeutic options are limited in OCCC due to the low frequency of *BRCA1/BRCA2* mutations in these cancers [53]. Consequently, a great deal of attention has been paid to synthetic lethal therapies that target vulnerabilities conferred by ARID1A deficiency. Like the BRCA1 and BRCA2 proteins, BAF250A/ARID1A promotes homologous recombination-mediated repair of DNA double-strand breaks, suggesting that PARP inhibition might be therapeutically effective [14]. Clinical trials of the PARP inhibitors olaparib and niraparib, using ARID1A deficiency as a biomarker, are underway in ovarian and other cancers (NCT04065269, NCT04042831, NCT03207347) (Table 1). 

The SWI/SNF complex and another chromatin remodeling complex, polycomb repressive complex 2 (PRC2), work antagonistically during transcription. EZH2 serves as the catalytic subunit in the PRC2 complex and mediates gene silencing. Dysfunction of the SWI/SNF complex due to ARID1A-deficiency leads to predominance of PRC2 activity in cancer cells [54,55]. In line with this, the therapeutic potential of EZH2 inhibitors against ARID1A-deficient cancers has been demonstrated [56]. The efficacy of an EZH2 inhibitor tazemetostat [57], which has been approved by the US FDA for the treatment of epithelioid sarcoma, is being tested against ovarian endometrial cancer, ovarian clear cell carcinoma, and endometrial cancer in an ongoing clinical trial (NCT03348631), again using ARID1A deficiency as a biomarker. In a Phase II clinical trial, tazemetostat exhibited an objective response rate of 69% in follicular lymphoma with activating EZH2 mutations; severe adverse events, such as thrombocytopenia, neutropenia, and anemia, were observed only in a small subset of cases [58]. Therefore, tazemetostat is a promising drug for the treatment of ARID1A-deficient OCCC. 

In addition, several other genes, including *ATR*, *HDAC2*, *BRD2,* and *HDAC6*, have synthetic lethal relationships with *ARID1A* [59,60,61,62]. Inhibitors of the products of these genes have already been approved for several non-ovarian cancers. For instance, multiple HDAC inhibitors have been approved by the FDA for cutaneous/peripheral T-cell lymphoma and multiple myeloma [63,64]. Among them, vorinostat, romidepsin, and belinostat have been investigated in clinical trials for epithelial ovarian cancer (Table 2). Trials of the vorinostat treatment combined with cytotoxic drugs were discontinued due to severe hematologic toxicity and gastrointestinal toxicity [65,66]. In a Phase II trial of recurrent platinum-refractory ovarian cancer, single treatment with vorinostat did not yield an evident response, although the drug was well tolerated [67]. Belinostat was also well tolerated in a combination regimen with paclitaxel and carboplatin [68]. The therapeutic efficacy of HDAC inhibitors against OCCC with ARID1A deficiency should be investigated in the future. 

We recently reported that ARID1A deficiency is associated with reduced metabolism of the antioxidant glutathione (GSH) [15]; consistent with this, ARID1A-deficient OCCC cells are sensitive to GSH inhibitors such as the investigational drugs APR-246 and buthionine sulfoximine (BSO). APR-246 was originally developed as a reactivator of mutant TP53 protein and is currently in Phase Ib/II clinical trials for hematological tumors (NCT04214860, NCT03931291) [69]. BSO was previously examined in a Phase I trial for melanoma and neuroblastoma (NCT00002730, NCT00005835), but it is not involved in any active clinical trials at the moment. Clinical trials of these inhibitors for OCCC would be worth undertaking.

### 4.3. Immune Checkpoint Blockade Therapy Exploiting the High Mutational Burden of ARID1A-Deficient Tumors

Tumors with the deficient mismatch repair (dMMR) phenotype respond well to immune checkpoint blockade therapy, as these tumors express many neo-antigens associated with high mutational burden [70]. BAF250A/ARID1A protein interacts with the MMR protein MSH2 and promotes MMR. Therefore, ARID1A deficiency might be an indicator of the dMMR phenotype, which is linked to the efficacy of immune checkpoint blockade therapy. The dMMR phenotype is observed in 3–14% of OCCC cases [71,72,73,74,75], whereas the relationship between ARID1A deficiency and dMMR is unclear. The number of OCCC patients enrolled in previous clinical trials of immune checkpoint inhibitors is very limited; therefore, the efficacy of therapeutic agents against OCCC remains unclear [76,77,78]. Notably, a clinical trial of the immune checkpoint inhibitor pembrolizumab, using ARID1A deficiency as a biomarker, is currently underway (NCT0461139); therefore, the proof of concept (POC) will be clarified in vivo in the near future.

## 5. Future Directions

At present, precision medicine for OCCC using SWI/SNF chromatin remodeling deficiency has not been implemented in daily oncology. Table 1 shows the status of FDA approval and clinical trials for drugs that could be effective against ARID1A-deficient OCCC. Several drugs, such as olaparib, niraparib, tazemetostat, and pembrolizumab, are being tested for their efficacy in clinical trials using ARID1A deficiency as a biomarker. Therefore, the POC obtained in preclinical studies should be validated in vivo in the near future. Other drugs, such as eleclomol, a ROS inducer (NCT00888615) [79], and berzosertib, an ATR inhibitor (NCT02627443), are being tested for efficacy in clinical trials that are enrolling ovarian cancer patients irrespective of *ARID1A* status. Analysis of the association between clinical response and ARID1A deficiency among studied cases might help to obtain further POC.

The list of target molecules proposed for cancer therapy is expanding day by day. Therefore, it is quite important to consider how discoveries made in preclinical models are translated to clinical trials designed to test whether modulating the activity of specific targets leads to a clinical response [80]. In particular, with respect to OCCC (which is a rare cancer), application of treatment modalities that are either established or are being tested on other major types of cancer would be a way forward. In fact, therapeutic strategies using PI3K-AKT inhibitors combined with HAT and BET (bromodomain and extra-terminal domain) inhibitors are considered strong candidates [81,82].

On the other hand, however, to understand specific/preferential properties of OCCC conferred by gene alterations is also quite important; sensitivity to GSH inhibitors conferred by ARID1A deficiency is much higher in OCCC than other types of cancers, such as gastric cancer [15,83], indicating that biological effects of gene alterations commonly observed in a variety of cancers are, in fact, largely different by cancer types. Therefore, preclinical studies focusing OCCC should also be intensively performed to establish truly feasible and efficient precision medicine of this disease. For this purpose, sharing data and materials of OCCC, which are unfortunately much less than those of many other common cancers, are highly inevitable. To facilitate preclinical studies OCCC, we dare to introduce here that human OCCC cell lines available for research are summarized in a report [84] and their pan-genome/transcriptome profiles are published [85,86]. In addition, patient-derived xenograft models of OCCC, which will give us accurate and specific insights of this disease, are available [87]. We hope that novel concepts of precision OCCC medicine will be produced here and will surely improve the present miserable situation of this disease in the near future.

## Figures and Tables

**Figure 1 cancers-13-01769-f001:**
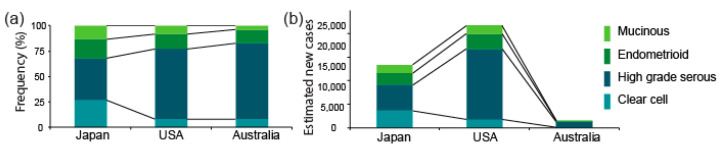
Proportions of OCCC and estimated numbers of new OCCC cases per year. (**a**) Proportions of OCCC among all epithelial ovarian cancers in Japan, the US, and Australia, using data obtained respectively from the Cancer Information Service, “Projected cancer incidence in 2020” Available online: https://ganjoho.jp/reg_stat/statistics/stat/short_pred.html (accessed on 19 November 2020); American Cancer Society, “Cancer Facts & Figures 2020”. Available online: https://www.cancer.org/research/cancer-facts-statistics/all-cancer-facts-figures/cancer-facts-figures-2020.html (accessed on 19 November 2020) and Cancer Australia, “Ovarian cancer statistics in Australia”. Available online: https://www.canceraustralia.gov.au/affected-cancer/cancer-types/ovarian-cancer/ovarian-cancer-statistics-australia (accessed on 19 November 2020) (**b**) Estimated numbers of new OCCC cases per year in Japan, the US, and Australia. The numbers were estimated based on the projected numbers of ovarian cancer patients and the proportions of OCCC in (**a**).

**Figure 2 cancers-13-01769-f002:**
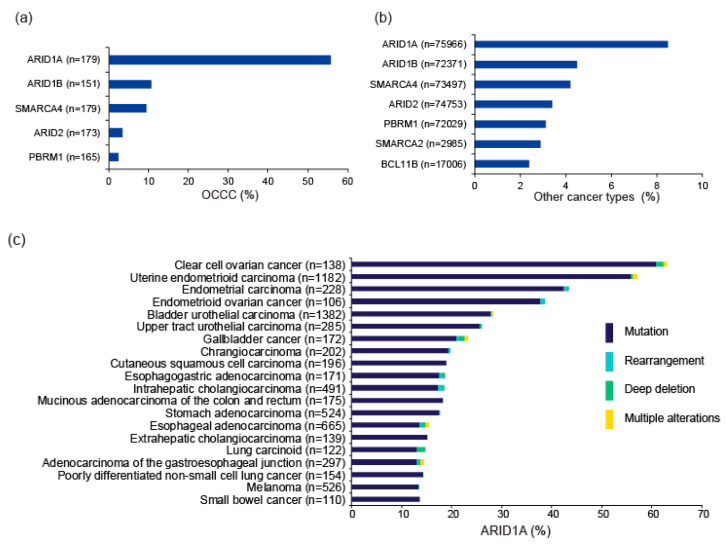
Mutations in genes encoding SWI/SNF chromatin remodeling factors. Mutation frequencies in OCCC (**a**) and other cancer types (**b**) in the GENIE Cohort v8.1 are shown. n: number of profiled samples. Top 20 cancer types according to mutation frequency of *ARID1A* (**c**), *ARID1B* (**d**), *ARID2* (**e**), *PBRM1* (**f**), and *SMARCA* (**g**) are shown. Cancer types detailed with >100 cases in the GENIE Cohort v8.1 are also shown. The four types of gene alteration are color-coded.

**Figure 3 cancers-13-01769-f003:**
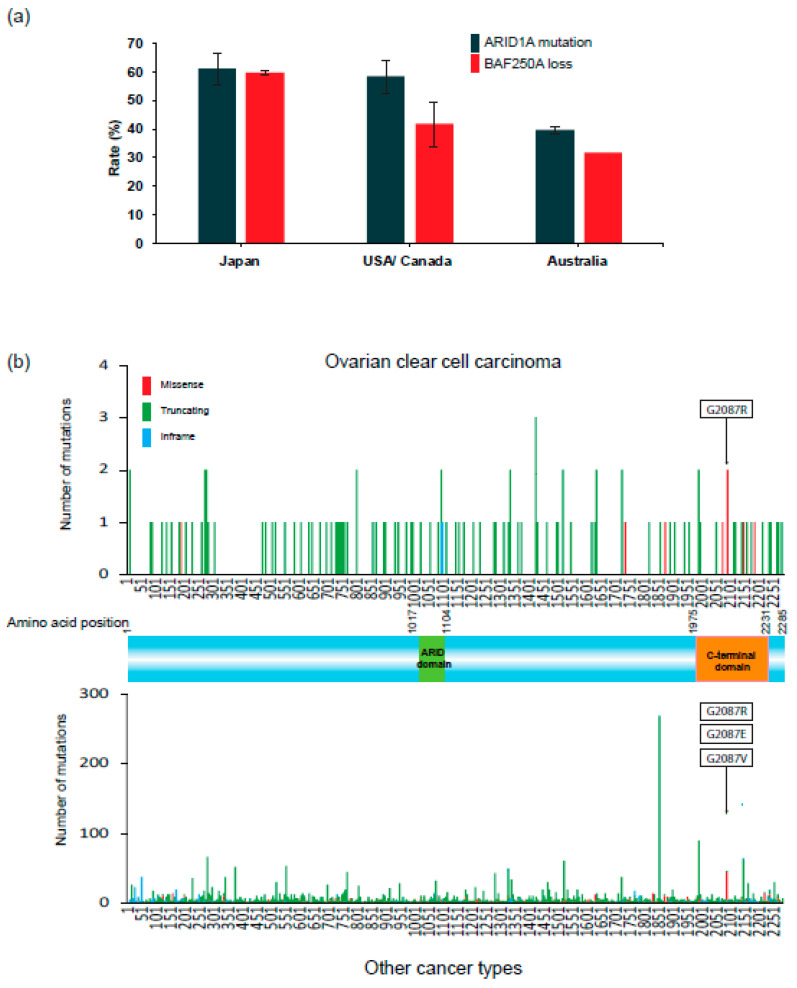
ARID1A alterations in OCCC. (**a**) Frequency of *ARID1A* mutations and loss of BAF250A/ARID1A protein expression in OCCC in Japan, the US, Canada, and Australia. Frequencies in published results [12,13,32,33,34,35,36,37] are expressed as the mean ± standard error. (**b**) Distribution of *ARID1A* mutants on the wild-type BAF250A/ARID1A protein sequence (GENIE Cohort v8.1). Upper and lower panels show mutations in OCCC (127 mutations) and other cancers, respectively. Mutations are colored by the type of mutations: missense, truncating (7114 mutations: nonsense, frameshift, and splice site mutations), or in-frame. Deleterious missense mutations at codon 2087 are indicated by boxes.

**Table 1 cancers-13-01769-t001:** ARID1A-target therapy.

Theraputic Targets	Drug	Clinical Trial Biomarker		Development Grade
		*ARID1A* Mutation	BAF250A Loss	
**Conventional chemothrapy**				
Ribonucleoside reductase	Gemcitabine	-	-	A
**Synthetic lethal therapy**				
GSH	APR-246	-	-	D
GCLC	Buthionine sulfoximine (BSO)	-		D
Induced ROS accumulation	Elesclomol	-	-	D
EZH2	GSK2816126	-	-	D
	Tazemetostat	NCT03348631	NCT03348631	B,C
	CPI-1205	-	-	D
	SHR2554	-	-	D
HDAC2	Vorinostat (SAHA)	-	-	B
HDAC6	Ricolinostat	-	-	D
ARID1B	-	-	-	-
BRD2	I-BET-762	-	-	D
PARP	Olaparib	NCT04042831	NCT04065269	A, C
	Niraparib	NCT03207347	-	A, C
ATR	Berzosertib	-	-	D
YES1	Dasatinib	NCT02059265, NCT04284202	NCT02059265	C
**Immunotherapy**				
PD-1	Nivolumab	-	-	B
	Pembrolizumab	NCT04611139	-	B

A: FDA-approved for ovarian cancer, B: FDA-approved for other cancer, C: Clinical trial underway for ovarian cancer using ARID1A as a biomarker, D: Clinical trials underway for ovarian cancer or other cancer without using ARID1A as a biomarker. Information on the clinical trials can be obtained at https://www.clinicaltrials.gov/ (accessed on 19 November 2020).

**Table 2 cancers-13-01769-t002:** Clinical trials of FDA approved HDAC inhibitors in ovarian cancer.

Theraputic Targets	Drug	Clinical Phase	Combination Regimens	Cancer Type	ClinicalTrials.gov Identifier	Recruitment Status
HDAC	Volinostat	I/II	Pac, Carbo	Recurrent EOC	NCT00772798	Unknown
		I/II	Pac, Carbo	Stage III/IV EOC	NCT00976183	Terminated (toxicities)
		Ib/II	Gem, Carbo	Platinum-sensitive recurrent EOC	NCT00910000	Terminated (toxicities)
		II	-	Recurrent EOC	NCT00132067	Completed
		I	-	Advanced ST and HM	NCT00045006	Completed
	Romidepsin	II	-	Recurrent EOC	NCT00091195	Terminated
	Belinostat	I/Ib	Ribociclib	Metastatic TNBC and recurrent EOC	NCT04315233	Recruiting
		II	Pac, Carbo	Recurrent EOC and BC	NCT00421889	Completed
		II	-	Recurrent EOC and BLM	NCT00301756	Completed
		II	Carbo	Recurrent EOC	NCT00993616	Completed
		I	-	Advanced ST	NCT00413075	Completed
		I	5-FU	Advanced ST	NCT00413322	Completed

EOC: Epitherial ovarian cancer, BLM: Borderline ovarian tumor, ST:Solid tumors, HM: Hematologic malignancies; TNBC: Triple negative breast cancer, BC: Bladder cancer; Pac: Paclitaxel, Carbo: Carboplatin, Gem: Gemcitabine, 5-FU: 5-Fluoruracil.

## Data Availability

Not applicable.

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
