# Peer review of "Treatment Strategies for ARID1A-Deficient Ovarian Clear Cell Carcinoma"

_cancers, 2021, doi:10.3390/cancers13081769_

Round 1

Reviewer 1 Report

The authors provide an overview of the potential for targeted thérapies for ARID1A mutated CCOC.  This is clearly an area of unmet medical need especially in Asia where the prevalence of this histology is more important.

Fig 1:  the labeling must be wrong.  The most frequent is endometrioid, should be HGSOC, and mucinous is probably CCOC???  Please check

Line 186:  Could the authors explain in 1 or 2 sentences why EZH2 targeting would be synthetically lethal in ARID1A mutated tumors

lines 222-225: can the authors give data on the prevalence of MMRd phenotype in CCOC to strengthen their hypothesis?  Also provide data on response to ICI among the subset of CCOC pts enrolled in clinical trials to date.

Reviewer 2 Report

Comments & Suggestions:

“ARID1A mutations, most of which are deleterious, are detected in about 50% of OCCCs, and loss of BAF250A/ARID1A protein, which functions as a regulatory subunit of the SWI/SNF complex, is observed at a similar frequency [12, 13].”

The authors might to include the interesting fact that all ARID1A mutation and truncating frameshifts and that usually the mutation(s) occur in only one allele of ARID1A suggesting that expression of ARID1A is from a single allele thus haploinsufficient.

Therefore, the function of the SWI/SNF complex is lost or compromised in half of OCCC cases.” I’m not sure this statement is correct. Is there good evidence that the SWI/SNF complex is lost in ARID1A deficient cells? If so, this should be referenced. What is the phenotype of ARID1A null mice, for example?

“Evidently, frequent ARID1A mutation is a genetic property of OCCC. ARID1A mutations are detected in adjacent endometriotic lesions of OCCC but not in distant endometriosis in the same patient, indicating that OCCC arises from endometriosis and that ARID1A functions as a major tumor suppressor in the development of OCCC [13].” This is a very important statement as it speaks to the cell of origin of OCCC so I would suggest that the authors include other corroborating references (PMID: 31981507, 25692284, 28489996).

Therefore, deficiency of SWI/SNF chromatin remodeling contributes to ovarian carcinogenesis in a tumor type-dependent manner.

Would the authors wish to mention that OCCC is a misnomer given that the cell of origin is an endometrial epithelial cell mis-located through retrograde menstruation?

ARID1A is also frequently mutated in uterine endometrial and endometrioid ovarian cancers; therefore, aberrations in this gene are likely to be frequently involved in malig-nancies of the female genital tract. In addition, SMARCA4, which is frequently mutated in poorly differentiated non-small cell lung cancer (Figure 2-g), is also mutated in a subset (about 10%) of OCCC.

My recollection is that ARID1A is the 4th most mutated gene in a recent pan-cancer analysis of gene mutations (PMID: 24183448). Perhaps this is a more accurate statement that the authors might consider and reference to provide greater impact to the statement. PMID: 23644491 may be a useful reference for this(?) Fig. 2C certainly implies that ARID1A is very frequently mutated in some way in a variety of cancer sites.

Not sure I understand the utility of Fig. 3A. It is well-established in the literature from direct gene analysis that ARID1A mutations result in loss of the gene product. Fig. 3a calls this into question and causes confusion around previous statements in the manuscript. Moreover, IHC results are inherently less reliable that direct ARID1A gene sequence data.

“The pathogenic roles of ARID1A deficiency in OCCC development remain largely unclear.”

I think the authors should cite the recent elegant publication from the Chandler and Fazleabas labs (PMID: 33176148) showing how loss or reduction in ARID1A expression dramatically alters the epigenome in a pre-OCCC model system (12Z cell line). As well as PMID: 31391455 & PMID: 33075803. The authors might also consider describing the effect of loss of ARID1A expression has on immune surveillance of OCCC i.e., https://www.jci.org/articles/view/134402.

Fig. 3B x-axis numbers are overlapping and unreadable. I would suggest using a direct download from the cBioPortal website for that data regarding sites of ARID1A mutation. The cBioPortal oncoprint is much more reader-friendly.

The authors should provide the mechanism of action for gemcitabine.

(2) Synthetic lethal therapy targeting vulnerabilities conferred by ARID1A deficiency.

I am surprised that the authors don’t expand this section to include the work from Caumanns et al, PMID: 33596421, and PMID: 33509905. These recent pre-clinical studies show promising new avenues of treatment modalities for OCCC. After all, the authors plug their own work suggesting that GSH inhibitors may be a possible new investigational OCCC drug and lament the lack of clinical trials testing this approach.

Overall, the review is well-written and easy to comprehend but I wonder if the authors have left out an important aspect of the OCCC pre-research state. I am referring to the pre-clinical models of OCCC that are essential to initial studies looking for unique treatment options for this hard to treat cancer.  One of the big issues with studying this cancer is the paucity of cell lines which represent the entire spectrum of OCCC cases i.e., those with low CNV and with mutations in the major players and those with high CNV but no mutations in the usual OCCC mutated genes. This is a big problem as most of the commercial cell line repositories don’t carry many of the unique OCCC lines, in fact, many have been published but never used again because they are not made available to the scientific community. This needs to change and calling attention to issue would be a unique aspect of this review. Franklin et al. in their recent paper on the NUCOLL43 cell line include an EXCEL document which lists almost all human OCCC cell lines including their availability, etc. Perhaps that paper should be cited(?) Also recent papers from Tan et al. (PMID: 31761620) and Papp et al. (PMID: 30485824) provide comprehensive genomic analyses of some of the available OCCC cell lines which is very useful to the research community and citing these papers would aid in publicizing their utility and showing that endometriod histotype is remarkably similar to OCCC. How this information might inform treatment strategies would be interesting for speculation. Reviews are also meant to be thought provoking and the authors here, haven’t really done that. I would encourage the authors to be much more daring in the way they have constructed this review and ask questions regarding how pre-clinical research into OCCC could be stimulated by access to the cell lines and showing similarities between OCCC and other cancer sites which show ARID1A mutations at high frequency. For example, could treatment modalities at those other cancer sites form the basis for further investigation in OCCC(?) What about the generation of syngeneic murine OCCC cell lines from mouse models of OCCC from the Chandler lab and others as new ways to decipher the immune response to OCCC and test novel agents(?) The authors are encouraged to be much more thought-provoking in their approach to this review given that the literature is replete with reviews on OCCC.

Round 2

Reviewer 2 Report

“Several studies reported that artificial loss of ARID1A impairs transcriptional and DNA repair activities within cells [14, 15]; therefore, the function of the SWI/SNF complex is (at least partially) lost in half of OCCC cases.”

“artificial” is the wrong word to use in this sentence. Are the authors referring to knockdown or knockout of ARID1A in an experimental model system or are they referring to loss due to somatic mutation in tumour cells?  If so, then it should be specifically stated as so.

The epigenetic role of ARID1A was also revealed in a pre-OCCC model system: ARID1A prevents super-enhancer hyperactivation, which leads to invasion of endometrial epithelial cells [46].

I would suggest rephrasing this sentence: which leads to enhanced migratory properties exhibited by pre-OCCC cells 

“Thus, ARID1A and other SWI/SNF factors are likely to function epigenetically as a tumor suppressor for ovarian carcinogenesis.”

The use of “ovarian carcinogenesis” in this sentence is awkward given that we know that the cell of origin of OCCC is not the ovary. The authors should rephrase this sentence.

“Sharing data and materials are (essential) inevitable if we are to expand perform preclinical studies of OCCC efficiently. We would like to state here that almost All of the human OCCC cell lines available for research are summarized in a previous report [83], and that detailed omics data of these cell lines have already been obtained [84, 85]. Patient-derived xenograft models of OCCC have also been established [86].” 

This section could be made more impactful and ends without an appropriate concluding sentence indicating why these statements are included in the text.
